# Impact of Grass Pea Sweet Miso Incorporation in Vegan Emulsions: Rheological, Nutritional and Bioactive Properties

**DOI:** 10.3390/foods12071362

**Published:** 2023-03-23

**Authors:** Sara Simões, Cecilio Carrera Sanchez, Albano Joel Santos, Diogo Figueira, Catarina Prista, Anabela Raymundo

**Affiliations:** 1LEAF—Linking Landscape, Environment, Agriculture and Food Research Center, Associated Laboratory TERRA, Instituto Superior de Agronomia, Universidade de Lisboa, Tapada da Ajuda, 1349-017 Lisboa, Portugal; sfrsimoes@isa.ulisboa.pt (S.S.); joelsantos@isa.ulisboa.pt (A.J.S.); cprista@isa.ulisboa.pt (C.P.); 2Departamento de Ingeniería Química, Escuela Politécnica Superior, Universidad de Sevilla, Calle Virgen de África, 7, 41011 Sevilla, Spain; cecilio@us.es; 3Mendes Gonçalves SA, Zona Industrial, lote 6, 2154-909 Golegã, Portugal; diogo.figueira@casamg.pt

**Keywords:** grass pea sweet miso, *Lathyrus sativus*, fermented foods, emulsions, clean-label, vegan product

## Abstract

Grass pea (*Lathyrus sativus* L.) is a pulse with historical importance in Portugal, but that was forgotten over time. Previous to this work, an innovative miso was developed to increase grass pea usage and consumption, using fermentation as a tool to extol this ingredient. Our work’s goal was to develop a new vegan emulsion with added value, using grass pea sweet miso as a clean-label ingredient, aligned with the most recent consumer trends. For this, a multidisciplinary approach with microbiological, rheological and chemical methods was followed. Grass pea sweet miso characterization revealed a promising ingredient in comparison with soybean miso, namely for its low fat and sodium chloride content and higher content in antioxidant potential. Furthermore, in vitro antimicrobial activity assays showed potential as a preservation supporting agent. After grass pea sweet miso characterization, five formulations with 5–15% (*w*/*w*) of miso were tested, with a vegan emulsion similar to mayonnaise as standard. The most promising formulation, 7.5% (*w*/*w*) miso, presented adequate rheological properties, texture profile and fairly good stability, presenting a unimodal droplet size distribution and stable backscattering profile. The addition of 0.1% (*w*/*w*) psyllium husk, a fiber with great water-intake capacity, solved the undesirable release of exudate from the emulsion, as observed on the backscattering results. Furthermore, the final product presented a significantly higher content of phenolic compounds and antioxidant activity in comparison with the standard vegan emulsion.

## 1. Introduction

Critical problems such as global population growth and climate change are, currently, the main drivers for new product development [1]. While modern consumers are aware of these problems and recognize the necessity of changing their consumption habits to lessen the sustainability crisis, they still demand new products that surprise and indulge their senses [2]. Thus, one of the biggest challenges for the food industry is to make the plant-based and sustainability transition, keeping in mind the need for attractive new products that are as appealing as they are sustainable.

Asian cuisine is a model to follow regarding diversity, including a range of components from the most common fruits, vegetables, pulses and animal-proteins, to more unconventional nutrient sources such as insects and fermented foods [3,4,5]. These alternative nutrient sources increase the sustainability of Eastern diets, as populations do not rely on such a limited list of foods as in Western countries [6].

Fermentation is a time honored preservation method, that has been used since the dawn of civilization as a way to increase safety while improving the nutritional and sensory aspects of foods [7]. Several studies have highlighted the beneficial effects of fermentation on consumers’ health, with valuable compounds being produced during the process, but also for the reduction of undesirable substances, such as naturally occurring toxins or anti-nutritional factors [8]. Miso is a Japanese fermented paste, usually made from fermented soybeans, that is used as a condiment and to prepare traditional miso soup [9]. For producing miso, soaked and boiled soybeans are added to water and salt, and are inoculated with an older miso (as a starter, such as happens for sour bread, for example) and koji (usually soybeans, rice or barley previously inoculated with *Aspergillus* sp.) [10]. Miso is a very popular fermented paste amongst Eastern cuisine, as it is said to be the source of longevity for Asian people [11]. Even though miso contains high minimum amounts of salt in its composition [10,12], the consumption of this paste has been associated with lower incidence of cardiovascular problems, lower blood pressure and lower incidence of stroke when compared with the consumption of equivalent amounts of sodium chloride [10,12]. These properties are associated with this fermentation process, as Nozue et al. (2017) [13] showed that the consumption of non-fermented soybeans does not present the same effect.

*Lathyrus sativus*, commonly known as grass pea, white pea [14], chickling vetch, Indian vetch or khesari [15], is a pulse cultivated in Portugal, particularly in the northeastern region of Planalto Mirandês [14] and Center-West [16], with a history of consumption in many European, African and South Asian countries [17]. Even though grass pea is very rich in nutrients, being suitable for animal feed and also as human food, with seeds containing 18–34% of protein and polyunsaturated fatty acids (58% of total fatty acids) [17], this crop was forgotten with time because of the presence of neurotoxic amino acids related to neurolathyrism, among other factors. However, there has never been record of outbreaks in Portugal [18], most probably due to the existence of autochthonous seeds with lower toxicity [14], to correct culinary preparation and also because this pulse is not the dominant component of Portuguese diets, contrary to what happens in many Asian and African countries [15,17]. Hence, considering its quality and versatility, grass pea is now annually celebrated in a gastronomic festival in Alvaiázere, center of Portugal, to promote the resurgence of this traditional food [16].

Previous to this work, a grass pea sweet miso was developed to increase this legume’s usage and consumption, merging the benefits of both grass pea and fermented foods. A vegan emulsion, previously developed [19] to answer the plant-based food trend (including vegan population, i.e., individuals that completely exclude animal products from their diets, either for health, environment or ethical motivation [20,21]), was used as base of incorporation. A grass pea sweet miso emulsion would attend to not only the plant-based food trend, but also other emerging trends in sauces and dressings such as the clean label trend, with a lack of artificial colors or flavoring agents, and the search for healthy yet flavorful sauces that merge new sensory experiences and balanced diets, and are eco-friendly [22]. Our goal is to use this innovative fermented paste to develop a new vegan emulsion with added value, using grass pea sweet miso as a clean label ingredient, attending the most recent food trends [22].

## 2. Materials and Methods

### 2.1. Materials

The grass pea sweet miso was produced as described by Santos et al. (2021) [9]. Briefly, a grass pea paste obtained from properly washed, soaked and cooked grass peas was mixed with koji and sea salt (10:9:1 mass ratio). The mixture was packed tightly into salt-coated glass jars, leaving a 1.5 cm headspace, and the top was covered with salt. The containers were closed and incubated at 30 °C for 4 months. After this initial fermentation period, the containers were kept at room temperature (20 ± 1 °C) until further use.

Faba bean protein concentrate and lupin protein isolate were used as emulsion stabilizers, containing a minimum of 85% (*w*/*w*) of protein each. The proteins used as emulsion stabilizers were provided ready to use by the manufacturers.

The remaining ingredients for the production of the emulsions were obtained from regular Mendes Gonçalves suppliers and are widely available in the market as standard industrial ingredients: refined sunflower oil (as defined in Codex Alimentarius—Codex Stan 210-1999, Adopted in 1999. Revised and amended in 2019); lemon juice (concentrated by a factor of 7); 8–10% (*v*/*v*) acetic acid alcohol vinegar; and psyllium husk powder (95% purity and 250 µm particle size).

All the results obtained from the following methodologies were analyzed and compared with a previously optimized faba bean and lupin stabilized emulsion, with 65% (*w*/*w*) oil content [19] and 0.9% (*w*/*w*) of sodium chloride, considered as a basepoint for new product development, defined as vegan mayonnaise or standard.

### 2.2. Methods

#### 2.2.1. Grass Pea Sweet Miso Chemical Analysis

Total water content was determined gravimetrically using an oven at 105 °C (Binder BD 115) until a constant weight was obtained [23]. Total protein content was determined in a Dumas protein/nitrogen analyzer (VELP Scientific NDA 702 DUMAS Nitrogen Analyser—TCD detector, Usmate—Italy), according to the Dumas method [24,25]. Protein content was calculated by multiplying the total nitrogen content by a conversion factor of 6.25. Crude fat content was determined gravimetrically by Soxtec extraction (Soxtec System HT 1043/1046 extraction unit (Tecator AB, Höganäs, Sweden)) [26]. The determination of sodium chloride content was performed by ICP-OES (iCAP 7000 series, Thermo Scientific, Waltham, MA, USA) [27], by multiplying the total sodium content by a conversion factor of 2.54 [28]. Other minerals were also analyzed by this method. All the analyses were performed in triplicate.

#### 2.2.2. In Vitro Antimicrobial Activity of Grass Pea Sweet Miso

The in vitro antimicrobial activity of grass pea sweet miso was tested using the drop testing method [29,30,31]. Table 1 presents the tested microorganisms, available on ISA yeast and bacteria libraries.

Pathogenic and contaminant microorganisms’ ability to grow in the presence of miso was tested. Bacteria and yeast strains were grown until the mid-exponential phase. Suspensions diluted 10-fold up to 10^−6^ were prepared in 96-well plates, and 3 µL was spotted with replica platter on plates containing the medium of interest. The dilutions were performed so that the last dilution corresponded to 1–2 cells in 3 µL. 

In a preliminary test (Test I), miso-based media with different concentrations (1%, 5% and 10% *w*/*w*) were used to test the ability of different microorganisms to grow on miso without proper nutritive medium. In a second test (Test II), nutritious and adequate media for each microorganism were prepared with the same concentrations of miso as before. This second assay tested the in vitro inhibitory capacity, as inability to grow in an adequate medium with miso added to it revealed inhibition capacity from miso. For both assays, sodium chloride plates were tested to discard salt as an inhibitory factor for each concentration.

Plates were incubated at 25 °C for yeast, *L. plantarum* and *Bacillus cereus*, and 38 °C for remaining pathogens. All plates were photographed after 1, 3 and 7 days, and yeast plates were also photographed after 15 days of incubation. Plates with media adequate for each microorganism were used as control. All plates were made in duplicate.

#### 2.2.3. Emulsion Production

Batches of 100 g of emulsions were produced as described by Cabrita et al. (2022) [19], with some modifications. First, faba bean and lupin isolate proteins were hydrated in distilled water for 30 min under magnetic stirring at room temperature. Once the protein was dispersed and hydrated (total protein: 1.5% *w*/*w*), sunflower oil (65% *w*/*w*) was carefully poured under agitation in an Ultra Turrax T-25 (IKA, Germany) homogenizer at 9500–13,500 rpm for 10 min, according to previously optimized conditions for mayonnaise-like systems [32]. Different miso incorporation phases were tested, namely before protein hydration, after protein hydration and after emulsification (oil addition). Furthermore, a formulation with psyllium husk was tested. For this, psyllium was hydrated separately from the protein (in part of the water in the formulation) for the same 30 min, and added to the protein after hydration. The emulsion production was completed by adding the acidic components—vinegar and lemon juice concentrate. The emulsions were stored at 4 °C in cylindrical glass jars (62 mm diameter, 56 mm height) for 24 h before any measurement to stabilize the structure of the emulsion. The control emulsion consisted of a faba bean and lupin emulsion without the addition of grass pea sweet miso or psyllium husk.

#### 2.2.4. pH Monitoring

The pH of emulsions was determined using a digital pH meter consisting of an electrode (Broadley James Corporation, Irvine, CA, USA) and a potentiometer pHM82 (Merck, Darmstadt, Germany).

#### 2.2.5. Texture Measurements

The emulsions’ Texture Profile Analysis (TPA) was performed in a TA.XT plus texturometer (Stable Micro-Systems, Godalming, UK) with a load cell of 5 kg. The TPA was performed in a temperature-controlled room (20 ± 1 °C) and replicated six times. The samples were in a glass container with 5.5 cm height and 6 cm diameter, filled to 4.5 cm height. The probe penetrated the samples until a 15 mm depth using a Perspex cylindrical probe with 19 mm diameter at 1 mm/s. From the texturograms (force versus time), firmness and adhesiveness were calculated. The cohesiveness parameter was not discussed, as it is not adequate to compare these types of emulsions. Firmness (N) is the maximum force recorded in the first penetration cycle, and adhesiveness ( − N.s) is the negative area of the texturogram. These textural properties correlate sensory aspects, with firmness corresponding to the force required to compress the material between molar teeth during chewing, and adhesiveness representing the work required to remove the probe from the material [33].

#### 2.2.6. Rheology Measurements

Linear viscoelasticity and steady-state flow measurements were carried out in a controlled-stress rheometer on a Haake MARS III (Thermo Fisher Scientific, Waltham, MA, USA) equipped with a UTC Peltier for temperature control.

Small amplitude oscillatory shear (SAOS) measurements were performed using a cone-and-plate sensor system (35 mm, 2°) within the previously assessed linear viscoelastic region, and in a frequency range of 0.01–100 Hz. The mechanical spectrum (i.e., G′ and G″ as a function of frequency) and the loss tangent (tan δ = G″/G′) were obtained. The plateau modulus (G^0^_N_) was estimated as the value of G’ obtained for the minimum value of the loss tangent, as described in the literature [34].

Steady-state flow curves were performed using a 20 mm serrated parallel plate system (PP20). The shear rate ranged from 10^−8^ to 500 s^−1^. The viscosity versus shear rate curve was adjusted to the Williamson model (1) as applied by other authors [19,35], using the Origin 2019 software (OriginLab).
(1)$$\eta =\frac{\eta_{0}}{1+{\left(k\dot{\gamma }\right)}^{m}}$$
where *η*_0_ is the zero shear rate limiting viscosity at low shear rates (Pa.s); *k* is the consistency coefficient (Pa.s) and a dimensionless shear-thinning index; and *m* is the slope of the power-law shear-thinning region.

#### 2.2.7. Droplet Size Distribution

The droplet size distribution (DSD) was determined as described by Cabrita et al. (2022) [19], using a laser diffraction instrument (Mastersizer 2000; Malvern Instruments, Malvern, UK) at 20 °C. The droplet size distribution of an emulsion is strongly correlated with its stability and sensory properties [36]. The emulsions were diluted and homogenized with distilled water before DSD determination. To determine possible flocculation phenomena, DSD was also obtained in the presence of Sodium Dodecyl Sulphate (SDS) (ITW reagents) (1:1 diluted emulsion:SDS), a surfactant that disaggregates floccules formed in emulsions [37]. 

The values of Sauter diameter (d_3,2_) (2) and of De Brouckere Diameter (3) (d_4,3_) were calculated. While d_3,2_ expresses the mean diameter for most of the droplets, d_4,3_ is associated with modifications in particle size as the result of several destabilizing mechanisms, such as droplet aggregation [38].
(2)$$d_{3,2}=\frac{\sum n_{i}d_{i}{}^{3}}{\sum n_{i}d_{i}{}^{2}}$$
(3)$$d_{4,3}=\frac{\sum n_{i}d_{i}{}^{4}}{\sum n_{i}d_{i}{}^{3}}$$
where *n*_*i*_ is the number of droplets which have *d*_*i*_ diameter.

The span parameter (4) was calculated to analyze the DSD profiles. Higher span is indicative of higher polydispersity of droplets [39].
(4)$$Span=\frac{d\left(90\right)-d\left(10\right)}{d\left(50\right)}$$
where *d*(x) corresponds to the x volume percentile of droplets with diameters smaller or equal to these values (10, 50 and 90).

DSD was obtained after 1–5, 15 and 30 days of emulsion formation. The determinations were performed in triplicate.

#### 2.2.8. Backscattering

The stability of emulsions, specifically its tendency to form cremate, was assessed using a vertical scan analyzer Turbiscan MA 2000 (Formulaction, Toulouse, France) [38,40]. Backscattering measurements (BS) were performed at 20 °C with a pulsed light source (λ = 850 nm). For this analysis, cylindrical glass cells were filled with the emulsions up to approximately 80 mm length, and were stored at 4 °C between measurements. The profiles of BS from emulsions versus the length were plotted as a function of storage (1–5, 15 and 30 days of emulsion production).

#### 2.2.9. Microbial Stability of Grass Pea Sweet Miso Emulsions

The grass pea sweet miso emulsions were evaluated regarding their microbial stability. For that, standard microbial analyses (Table 2) were performed immediately after emulsion production, and 30 days later, with emulsions being stored in air-tight sterile plastic tubes at 4 °C and 28 °C (accelerated assay).

#### 2.2.10. Bioactivity Assays

The in vitro bioactivity of miso paste and emulsions were analyzed using spectrophotometric methodologies. For all the bioactivity studies, three replicates were performed for each extract.

To extract the phenolic and antioxidant compounds, samples were immersed in methanol (1:10 *w*/*v*) and stirred for 60 min at room temperature using a Reax 2 *w*/*o* adapter (Heidolph, Schwabach, Germany) with a rotation speed range of 3 rpm. After that, the samples were centrifuged at 12,000 rpm at 20 °C for 10 min (Sigma 1-14 Microfuge, St Louis, MO, USA), with the supernatant being used for the subsequent analysis. 

The Folin–Ciocalteu method was used to quantify the total phenolic compounds (TPC) in the samples, with gallic acid as standard [41,42] (results presented in gallic acid equivalents, GAE). Briefly, diluted extracts were homogenized with Folin’s reagent (1:12 *v*/*v*) and mixed in the vortex. After 3 min of reaction, sodium carbonate solution (7.5% *w*/*v*) was added (2:5 *v*/*v*), and the mixture was kept out of direct light contact for 2 h at 20 °C before measurement. TPC were measured in a Cary Series UV-Vis spectrophotometer (Agilent Technologies, Santa Clara, CA, USA) at 760 nm, using a plastic cuvette and deionized water as blank.

The antioxidant capacity of the samples was measured using two different methodologies, also using gallic acid as standard (results in GAE). The scavenging effect of extracts was determined using the DPPH (2,2-diphenyl-1-picrylhydrazyl-hydrate) method [42,43,44]. Extracts were mixed with a methanolic solution of DPPH (1:25 *v*/*v*) and kept out of direct light for 1 h at 20 °C before measurement. The mixture’s absorbance was measured in a spectrophotometer at 517 nm, using a quartz cuvette and methanol as blank. For each sample, a control with DPPH solution replaced by methanol was performed and measured. The reducing power of the extracts was determined using the ferric ion reducing antioxidant power (FRAP) assay [45]. FRAP solution was prepared by mixing 300 mM acetate buffer (pH 3.6), 20 mM ferric chloride and 10 mM 2,4,6-tripyridyl-s-triazine (TPTZ—10 mM TPTZ in 40 mM HCl) in a ratio of 10:1:1. The extracts were diluted with deionized water and TPTZ solution (1:9:90). The mixture was kept in a 37 °C bath for 30 min, after which its absorbance was measured at 595 nm, using a quartz cuvette and water as blank. For each sample, a control with FRAP solution replaced by deionized water was performed and measured.

#### 2.2.11. Statistical Analysis

GraphPad Prism Software (version 9.0) was used to perform the analysis of variance (ANOVA), using the t-Student test to compare two samples and the Tukey test to compare three or more samples. Both tests were performed with a 95% degree of confidence (α = 0.05). Results were considered significantly different when *p*-values were inferior to 0.05.

## 3. Results and Discussion

### 3.1. Grass Pea Sweet Miso Characterization

#### 3.1.1. Chemical Analysis

The results of the chemical analysis of grass pea sweet miso are presented in Table 3. The table compares the results obtained with the composition of soybean miso found in the literature [46,47,48].

Analyzing Table 3, the main advantage of grass pea sweet miso in comparison with traditional soybean miso is related to its low fat content, slightly lower sodium chloride content and considerably higher content in potentially antioxidant compounds. According to the literature [48], the antioxidant potential of traditional miso is at least 10-fold lower than that of grass pea miso. Hence, the utilization of grass pea not only promotes the consumption of a long-forgotten legume, but also grants more antioxidant potential to the final product and to future potential food formulations.

#### 3.1.2. In Vitro Antimicrobial Activity

The in vitro antimicrobial activity of grass pea sweet miso was determined in two test-phases: (i) Test I, to evaluate the ability of pathogenic and contaminant microorganisms to grow in the presence of miso; (ii) Test II, to test miso’s microbial growth inhibitory capacity when all the necessary nutrients are present (proper nutritive medium). Plates obtained after 7 or 14 days of incubation (stationary phase) for pathogenic and contaminant microorganisms are presented in Figure 1 and Figure 2, respectively.

On Test I, no growth of pathogens was observed in any concentration of miso or sodium chloride (Figure 1a), suggesting that neither miso nor sodium chloride is nutritive for pathogen growth on their own, which was already expected for NaCl. Therefore, the presence of miso in a formulation would not be a nutrient source for the growth and subsistence of possible pathogenic contaminant microorganisms.

On Test II (Figure 1b), *S. aureus*, *L. innocua*, *E. coli* and *Salmonella* grew poorly, with a clear effect of miso concentration. When miso concentration increased from 1.0 to 10.0% (*w*/*w*), a considerable negative effect on colonies’ growth was observed. The same was not observed for sodium chloride (plates with equivalent concentrations of salt as in miso plates), since it did not inhibit the growth of these pathogens in the presence of nutritive medium. For *B. cereus*, miso concentration had an inhibitory effect, with 10.0% (*w*/*w*) miso being completely effective in the inhibition of colony growth. However, *B. cereus* colonies were able to thrive in adequate medium plates added of equal concentrations of sodium chloride, discarding the effect of salt as the main inhibitory factor of pathogens’ growth in this situation and pointing to a clear inhibitory effect of miso. These results suggest an effect of miso’s bioactive compounds, which partially inhibit both Gram-positive and Gram-negative bacteria in the presence of an adequate growth medium.

Spoilage (non-pathogenic) microorganisms, typically found as contaminants in factories and raw materials, i.e., yeast and lactic acid bacteria [49], were also tested. As expected, in plates without nutritive medium (Test I), *L. plantarum* was able to grow with observable difficulties in comparison with the control (Figure 2a), most likely for lack of enough nutrients, as higher concentrations of miso enabled a mild growth of *L. plantarum* colonies. This was not the case for nutritive medium plates with both miso concentrations (Figure 2b), in which lactic acid bacteria growth was similar to that of the positive control without miso addition, although a very slight decrease on growth was perceived at 10.0% (*w*/*w*) of miso incorporation. Hence, it is concluded that the tested range of concentrations of grass pea sweet miso is not inhibitory of *Lactiplantibacillus plantarum* growth, although it does not constitute a great source of nutrients at lower concentrations (<10.0% *w*/*w*). Concerning yeasts, on both tests (Figure 2), colonies grew in miso and salt without any apparent difficulty: an exception to this was the case of *Zygosaccharomyces bailli* (one important spoilage yeast in fruit concentrates), which was progressively inhibited as miso concentrations increased, until total inhibition at 10.0% (*w*/*w*) miso (the same did not happen for salt; hence, its effect was discarded). These results were expected, as most of the microorganisms tested are autochthonous to miso [10,47].

Our results show that the incorporation of up to 10.0% (*w*/*w*) of grass pea sweet miso may not be used per se as an effective replacement for traditional preservatives, but can be used as an adjuvant. Moreover, higher concentrations present higher potential, as they inhibited *Bacillus cereus* (Figure 1b), as discussed before. Additionally, in order to evaluate the whole potential of miso as a food preservative, the effect of each matrix should also be tested and taken in to consideration.

### 3.2. Miso Incorporation

#### 3.2.1. Preliminary Tests—Incorporation Phase

Considering miso’s high content of sodium chloride, its incorporation phase was thoroughly studied using rheological and textural parameters to define the best incorporation phase. For this study, 5.0 and 7.5% (*w*/*w*) of miso were tested, incorporating before and after protein hydration, and after oil addition (after emulsion formation). After analyzing the results (unpublished data), the post-hydration phase was selected, as it enabled the production of emulsions with closer parameters to the vegan mayonnaise (standard and target regarding structure parameters).

#### 3.2.2. Incorporation Proportion and Formulation

Taking into consideration the results obtained in the preliminary assays, namely grass pea sweet miso’s sodium chloride content and miso’s ability to inhibit microbial growth, four different incorporation concentrations were tested. Emulsions produced with 5.0, 7.5, 10.0 and 15.0% (*w*/*w*) of grass pea sweet miso and standard emulsion are presented in Figure 3.

Emulsion 15% (*w*/*w*) presented the same amount of sodium chloride as the commercial standard considered. The emulsion with 10.0% (*w*/*w*) miso presented the target amount of sodium chloride, 1.0% (*w*/*w*). The remaining emulsions contained less than 1% (*w*/*w*) of sodium chloride, but after initial sensory analysis (pass/fail decisions based on like/dislike in simple taste tests, results unpublished) it was concluded that the umami flavor of miso balanced the overall taste of the emulsions, although they were less salty than the vegan mayonnaise. 

Furthermore, after developing miso emulsions, it was noticed that an exudate would be released from the emulsions, associated with the miso paste, that tended to emit exudate as well. Thus, different concentrations (0.05–0.25% (*w*/*w*)) of psyllium husk were added to the most promising emulsion, for its water up-taking properties [50,51,52,53]. From these prototypes, the concentration of 0.1% (*w*/*w*) was chosen, as it prevented the release of exudate without further alterations on the emulsion’s texture and mouthfeel (pass/fail decisions based on like/dislike in simple taste tests, results unpublished).

### 3.3. Miso Emulsion Characterization

#### 3.3.1. Textural Properties

Texture parameters obtained for the texture profile analysis (TPA) of standard and 5–15% (*w*/*w*) miso emulsions are presented in Figure 4a,b, which display the texture parameters obtained for the TPA of 7.5% (*w*/*w*) miso emulsions, with and without the addition of 0.1% (*w*/*w*) psyllium husk.

Figure 4a,b show that miso emulsions have significantly different texture parameters from the vegan mayonnaise, regardless of the concentration of incorporation. Concerning firmness, this parameter increased significantly with the concentration of miso (*p* < 0.05). In fact, increases of 5.0% (*w*/*w*) (5.0 to 10.0 to 15.0) generated significant increases in emulsion firmness (*p* < 0.05), which did not occur when the increase was lower (from 5.0 to 7.5%, *p* > 0.05). On the other hand, the different concentrations of miso did not generate significant alterations to the adhesiveness of emulsions (*p* > 0.05), which showed a tendency to increase, but not significantly, since an increment in 10.0% (*w*/*w*) of concentration from 5.0% (*w*/*w*) to 15.0% (*w*/*w*) of miso did not influence this parameter (*p* > 0.05).

Analyzing Figure 4c,d, the addition of psyllium husk decreased the firmness of the 7.5% (*w*/*w*) miso emulsion in comparison with the one without the addition of fiber (*p* < 0.05). This phenomenon has also occurred in wheat and gluten-free bread, with the incorporation of psyllium husk decreasing the firmness of the crumb [54,55,56]. The loss of firmness of the emulsion may have happened due to psyllium’s water intake ability, limiting the water available for protein activation, therefore decreasing protein’s structuring capacity [57]. The same trend was observed for the adhesiveness parameter, although not significantly (*p* > 0.05).

#### 3.3.2. Rheological Behavior

The mechanical spectra, i.e., the evolution of storage and loss moduli with frequency for standard and 5–15% (*w*/*w*) miso emulsions are presented on Figure 5.

Considering the spectra’s similar slopes and pattern, it is possible to conclude that all emulsions present similar viscoelastic properties in the experimental range of frequencies, with miso paste not displaying an evident effect on the emulsions’ mechanical properties. All the emulsions present G′ values higher than those of G″, with a slight dependence on frequency, most likely caused by the stabilizing system of faba and lupin, as shown previously by Cabrita et al. [19]. However, analyzing the structure parameters, namely G′ at 1 Hz and Plateau modulus (Table 4), miso does have an influence on the structure of the emulsions, increasing the structure parameters. This is also observable in Figure 5, as the standard emulsion presents lower G′ and G″, and 15% (*w*/*w*) emulsion presents the highest moduli.

Figure 6 shows the mechanical spectra of 7.5% (*w*/*w*) emulsions, with and without psyllium husk.

Analyzing both Figure 6 and Table 4, the de-structuring effect of psyllium husk is clear, as the addition of just 0.1% (*w*/*w*) of this fiber generated not only a significant decrease in the structure parameters (*p* < 0.05), but also a clear dependence on higher frequency values, indicating a destabilization of the emulsion’s structure. This phenomenon was not expected, as the addition of psyllium husk has been described as reinforcer of several food structures, such as pasta [58], gluten free bread [59] and cookie dough [57]. However, as observed with the decrease in texture parameters (Figure 4), adding psyllium husk may have an important impact on the hydration step, since part of the water (around 45%) normally used for protein hydration must be used for psyllium hydration, thus reducing the efficiency of protein hydration and activation, and consequently its ability to structure the emulsion. The poor hydration of vegetable proteins caused by psyllium has been described by Raymundo et al. (2014) in biscuit doughs with partial substitution of wheat flour by psyllium husk. The addition of psyllium compromised the development of the gluten network necessary in this type of product, since psyllium reduced the available water in the dough [57].

Figure 7 presents the flow curves, i.e., the variation of viscosity with the shear rate, of standard and miso emulsions.

By the analysis of Figure 7, every emulsion presents the same flow trend, presenting a shear-thinning behavior, with vegan mayonnaise standing out for an apparently lower limit viscosity value. However, considering the Williamson’s parameters presented in Table 5, only the emulsion 7.5% (*w*/*w*) differed significantly from the standard (*p* > 0.05) in terms of limit viscosity, with the rest of the miso emulsions not presenting a significantly different limit viscosity from the control (*p* < 0.05). Additionally, the tested miso concentrations did not have an effect on this parameter, as the limit viscosities did not change significantly within the tested range. The same behavior was found after the addition of psyllium to the 7.5% (*w*/*w*) emulsion, which did not present a significantly different limit viscosity from the 7.5% (*w*/*w*) emulsion (*p* < 0.05). Furthermore, the other Williamson’ parameters, namely k and m, did not change prominently for any of the emulsions. 

#### 3.3.3. Emulsions’ Stability

Emulsion stability was assessed based on the evaluation of the droplet size distribution (DSD). These measurements of the emulsions on days 1 and 30 after production are presented in Figure 8.

Analyzing the DSD profiles, it is possible to observe that most of the curves presented a unimodal distribution on the first day after emulsion production (Figure 8a). Nevertheless, the unimodal behavior decreased with the addition of miso paste, as curves tend to be lower and flatter. This behavior is confirmed by the span parameter (Figure 9), which significantly increased with miso concentration (*p* < 0.05), indicating increased polydispersity associated with miso paste addition. Regarding stability over shelf-life, the polydispersity of all emulsions stabilized after 30 days (Figure 8), with span values not being significantly different between the standard and 5–15% (*w*/*w*) emulsions. It is notable that emulsions 7.5 and 10% (*w*/*w*) did not change their span over the 30 days (*p* < 0.05), indicating high stability.

Another feature of the presented DSD profiles is the presence at 30 days of small populations of droplets at the largest diameters, especially in the emulsions with the highest miso content. This fact could be associated with the tendency of droplets to aggregate (to form flocs) over time. However, these smaller droplet populations are not very significant compared with the majority population of droplets.

Figure 10 and Figure 11, respectively, present d_4,3_ and d_3,2_ diameters of emulsions on days 1 and 30 after production.

De Brouckere diameter (or d_4,3_) is associated with modifications in particle size as the result of different destabilization processes, namely coalescence and fat droplet aggregation [38]. Observing Figure 10, emulsions 7.5 and 10% (*w*/*w*) did not present d_4,3_ values significantly lower from the standard (*p* < 0.05). Additionally, this parameter did not change significantly over 30 days for any of these emulsions (*p* < 0.05). On the other hand, d_4,3_ of emulsions with 5 and 15% (*w*/*w*) of miso significantly decreased over shelf-life (*p* > 0.05), dropping to values not very different from all the analyzed samples (*p* < 0.05).

Considering Sauter diameter values (d_3,2_) (Figure 11), traditionally used as representative average diameter for most droplets in emulsion characterization [38], with the exception of 10% (*w*/*w*) emulsion, all the emulsions decreased their average droplet diameters over storage (*p* > 0.05). Thus, it is possible to conclude that the addition of miso, overall, decreased the average droplet size (*p* > 0.05). Additionally, the studied emulsions tended to stabilize their droplet size after 30 days, as significant differences on day 1 (*p* > 0.05) are no longer observable on day 30 (*p* < 0.05).

All of these results indicate that the addition of miso to the standard emulsion does not significantly impact its stability against coalescence phenomena.

Figure 12 displays the Droplet Size Distribution of emulsions with and without psyllium husk, 1 (a) and 30 (b) days after emulsion preparation.

The DSD profiles of the emulsion with and without psyllium husk are distinct on day 1, as the curve for the psyllium emulsion displays a much lower peak (Figure 12a). Furthermore, this difference is confirmed by the span parameter (Figure 13), which is significantly higher in the psyllium emulsion (*p* > 0.05), indicating more polydispersity when compared with the emulsion without fiber addition. However, after 30 days, the polydispersity of both emulsions seem to stabilize (Figure 12b), with peaks being closer to each other and span values not significantly different (*p* < 0.05). This phenomenon was caused by Psyllium emulsion evolution over storage, as its span was significantly different between days 1 and 30 (*p* < 0.05), which did not happen for 7.5% (*w*/*w*) miso emulsion without psyllium.

Figure 14 and Figure 15 present De Brouckere (d_4,3_) and Sauter (d_3,2_) diameters, respectively.

Regarding d_4,3_, there was a significant increase in this parameter with the addition of psyllium husk (*p* > 0.05). Although this parameter is usually related to coalescence phenomena, in this case it is probably related to fiber aggregates (also observed in gel-type products [60]). The average droplet size (d_3,2_) decreased with the addition of psyllium (*p* > 0.05). Furthermore, the d_4,3_ of psyllium emulsion decreased (*p* > 0.05) after 30 days of storage, whereas the d_3,2_ increased (*p* > 0.05).

The nonvariation of the d_4,3_ diameter over time indicates that the presence of psyllium husk does not significantly affect the stability of the system, despite causing a de-structuring effect on it, as previously mentioned.

In order to study the existence of the flocculation during emulsification, DSD were obtained in the presence or absence of Sodium Dodecyl Sulphate (SDS) solution, a surfactant known to disaggregate floccules formed in emulsions. The DSD obtained were very similar (data not shown), indicating that these emulsified systems do not present flocculation.

Figure 16 shows the backscattering profile of standard and miso emulsions with and without psyllium husk. These measurements are also important to evaluate the emulsion stability and to identify the type of phenomena associated with stabilization processes. In general, all the emulsions do not present variations in the backscattering measurements throughout the tube or as a function of time. These results evince that the systems studied were fairly stable, with no appreciable clarification or creaming. Only in the measurement at 30 days can a marked decrease in the backscattering values be seen in all the systems in the upper part of the tube (Figure 16b). The latter can be associated with the appearance of an exudate released from the emulsions. As can be seen in Figure 16d, this exudate does not appear in the emulsion with psyllium husk, making it more stable than the formulation without psyllium (7.5% (*w*/*w*)). 

#### 3.3.4. Microbiological Stability

Microbiological analyses were conducted on grass pea sweet miso emulsions without any preservative addition. The analyses were performed immediately after emulsion production and 30 days later, after storage at 4 °C and 28 °C (accelerated assay). Table 6 reports the counts (CFU/g) for total mesophilic bacteria, fungi (yeasts and molds), lactic acid bacteria and *E. coli*, and the presence or absence of *L. monocytogenes* and *Salmonella* sp. on 25 g of emulsion.

Analysis of the results allows the conclusion that, as expected, there was no contamination with pathogenic bacteria; hence, these were not found on the samples on any assay. For the refrigeration assay, lactic acid bacteria and total mesophiles increased in number after 30 days of storage, but yeasts and molds had a 0.44 log reduction. However, on the accelerated assay every parameter decreased, with total mesophiles and lactic acid bacteria lowering to values under 10 CFU/g, and fungi reducing to under 100 CFU/g. This trend might be due to the depletion of nutrients. Further analysis, such as pH and rheological parameter measurements, would be necessary for more substantial conclusions.

#### 3.3.5. Bioactive Compounds

Figure 17 presents the total phenolic content and antioxidant activity of standard and 7.5% (*w*/*w*) miso emulsions.

The incorporation of just 7.5 g/100 g of grass pea sweet miso paste to the emulsion significantly improved all tested bioactivities (*p* > 0.05), with a boost of 279% on TPC, 100% in FRAP and 270% in DPPH. Hence, the addition of grass pea sweet miso to the emulsion formulation has a positive impact on the nutritional quality of the final product.

## 4. Conclusions

The incorporation of grass pea sweet miso in vegan emulsions was carried out, aiming to increase the use of grass pea in the Portuguese diet, as well as to develop innovative emulsions aligned with consumer trends.

In comparison with traditional soy bean miso, grass pea sweet miso presented lower content in fat, slightly lower sodium chloride content and ten times more antioxidant compounds, thus representing a very interesting alternative to the traditional product. An in vitro analysis of grass pea sweet miso’ antimicrobial capacity revealed a potential effect against *Bacillus cereus* at concentrations around 5% (*w*/*w*). Hence, five formulations of emulsions based on a standard vegan emulsion were tested. In total, 5–15% (*w*/*w*) miso emulsions presented different characteristics from the standard, namely higher adhesiveness and viscosity, and higher firmness and structure parameters dependent on miso concentration. DSD and backscattering results showed that the incorporation of miso did not have an effect on emulsion destabilization. A phenomenon of exudate release was empirically observed and instrumentally determined on the backscattering assay. This problem was solved with the incorporation of 0.1% (*w*/*w*) of psyllium husk, a fiber that not only increased the nutritional interest of the developed product, but also solved the release of exudate. This incorporation was made on the most promising miso concentration, 7.5% (*w*/*w*). Finally, microbial analysis showed that the emulsion with miso is not particularly prone to contaminant proliferation in accelerated assay conditions, hinting good stability.

In summary, an interdisciplinary approach allowed the incorporation of an innovative clean label ingredient, namely grass pea sweet miso, that can be used in emulsions to increase their taste, health benefits, structure and appeal to the innovation-eager consumer. 

## Figures and Tables

**Figure 1 foods-12-01362-f001:**
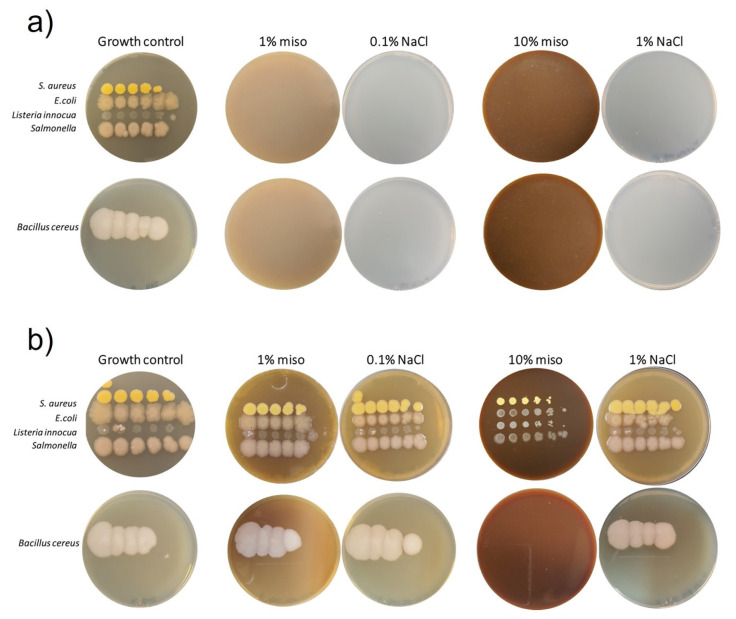
Plates resulting from in vitro antimicrobial activity determination for pathogenic microorganisms, incubated for 7 days (stationary phase) at adequate temperatures. Growth positive control plates in BHI medium are also shown. From left to right, drops from 10^−1^ to 10^−5^ dilutions. Miso concentrations of 1.0 and 10.0% (*w*/*w*). Sodium chloride agar plates with the concentration of salt equivalent to each miso plate. (**a**) Results for preliminary test, Test I, with no nutritive media, only different miso concentrations as substrate. (**b**) Results for Test II, with nutritive media and different miso concentrations.

**Figure 2 foods-12-01362-f002:**
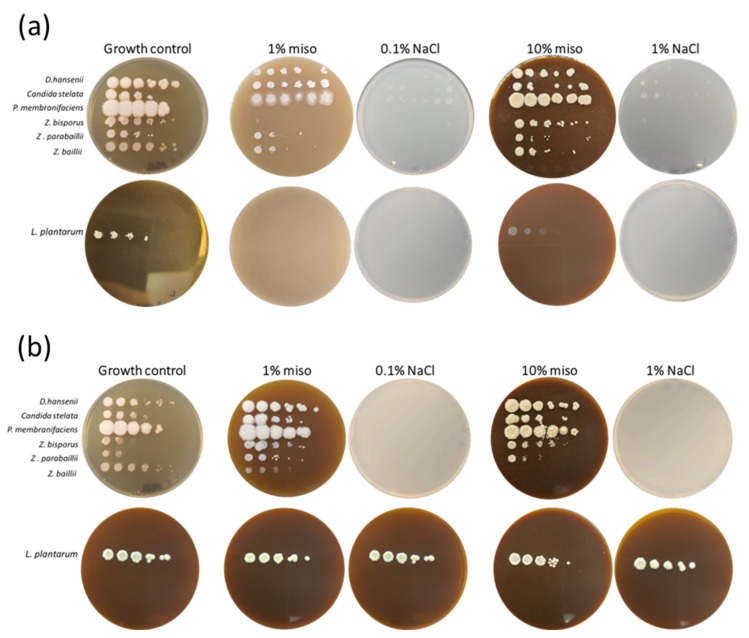
Plates resulting from in vitro antimicrobial activity evaluation tests for food contaminant microorganisms (yeast and *Lactiplantibacillus plantarum*) after 14 days of incubation. Positive control plates of YPDF medium for yeast, and of MRS for *L. plantarum*. From left to right, drops from 10^−1^ to 10^−5^ dilutions. Plates with miso concentrations of 1.0 and 10.0% (*w*/*w*) or with sodium chloride at equivalent concentrations of each miso plate. (**a**) Results for preliminary test, Test I, with different miso concentrations or sodium chloride and no nutritive media. (**b**) Results for Test II, with nutritive media and different miso concentrations or equivalent concentrations of sodium chloride.

**Figure 3 foods-12-01362-f003:**
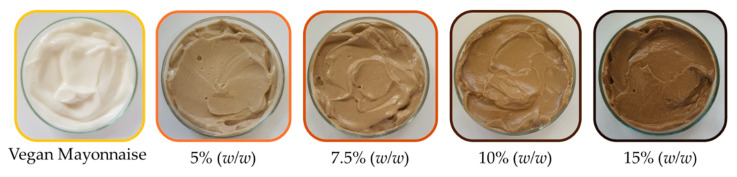
Appearance of emulsions containing 5.0 to 15.0% (*w*/*w*) of miso paste and the vegan mayonnaise standard.

**Figure 4 foods-12-01362-f004:**
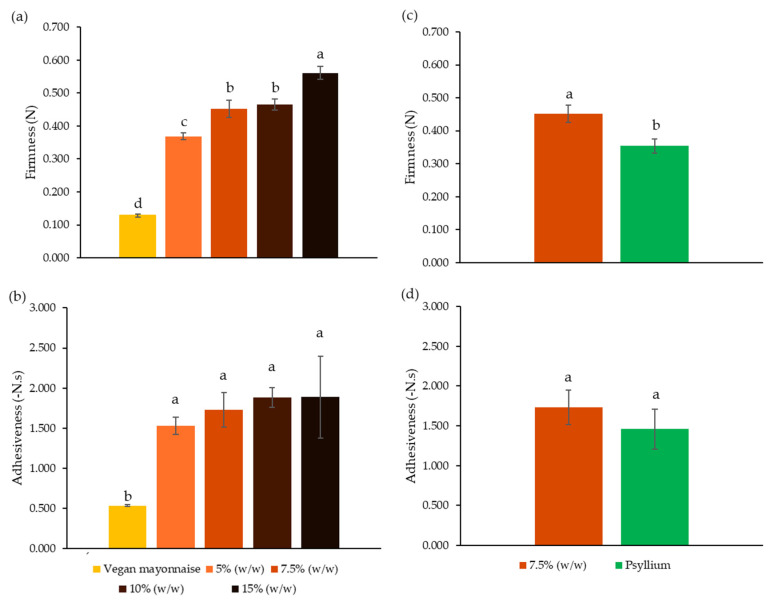
Firmness (**a**) and adhesiveness (**b**) of emulsions with 5, 7.5, 10 and 15% (*w*/*w*) of miso paste and standard emulsion. Firmness (**c**) and adhesiveness (**d**) of emulsions with 7.5% (*w*/*w*) of miso paste, one of them with 0.1% (*w*/*w*) of psyllium husk. Different letters indicate significantly different results (*p* < 0.05) according to Tukey test or *t*-test.

**Figure 5 foods-12-01362-f005:**
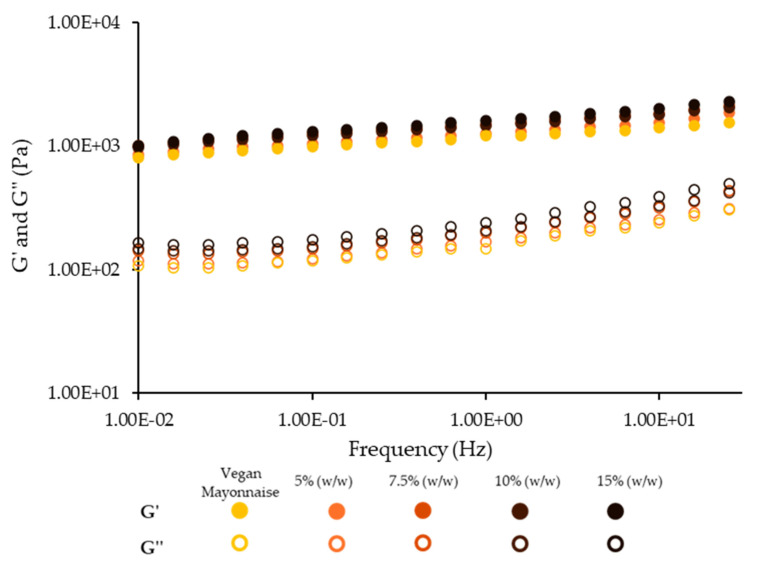
Mechanical spectra of emulsions with 5, 7.5, 10 and 15% (*w*/*w*) of miso paste and standard emulsion. G′ corresponds to the elastic modulus and G″ corresponds to the viscous modulus.

**Figure 6 foods-12-01362-f006:**
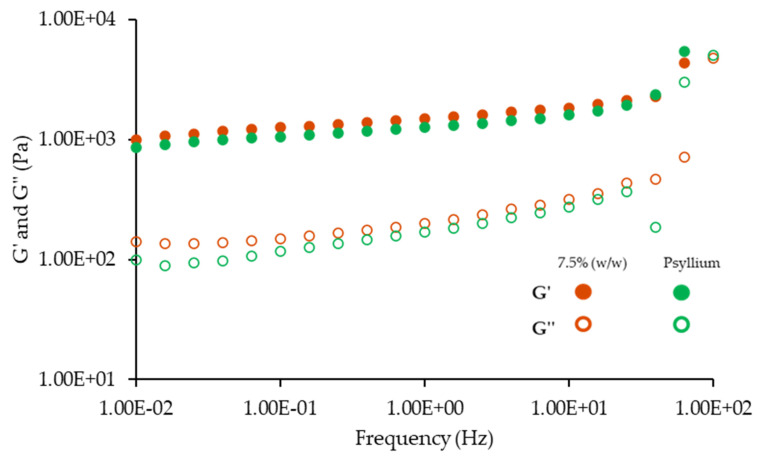
Mechanical spectra of emulsions with 7.5% (*w*/*w*) of miso paste, one of them with 0.1% (*w*/*w*) of psyllium husk. G′ corresponds to the elastic modulus and G″ corresponds to the viscous modulus.

**Figure 7 foods-12-01362-f007:**
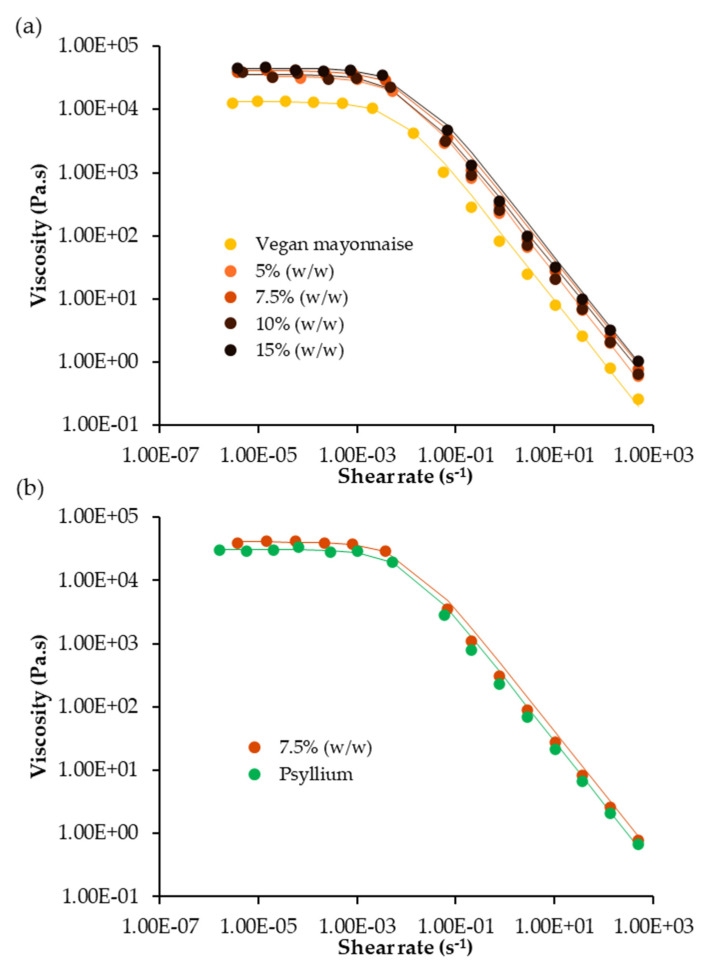
Flow curves of emulsions (**a**) standard and with 5, 7.5, 10 and 15% (*w*/*w*) of miso paste; (**b**) with 7.5% (*w*/*w*) of miso paste, one of them with 0.1% (*w*/*w*) of psyllium husk. Lines represent Williamson’s model adjustment.

**Figure 8 foods-12-01362-f008:**
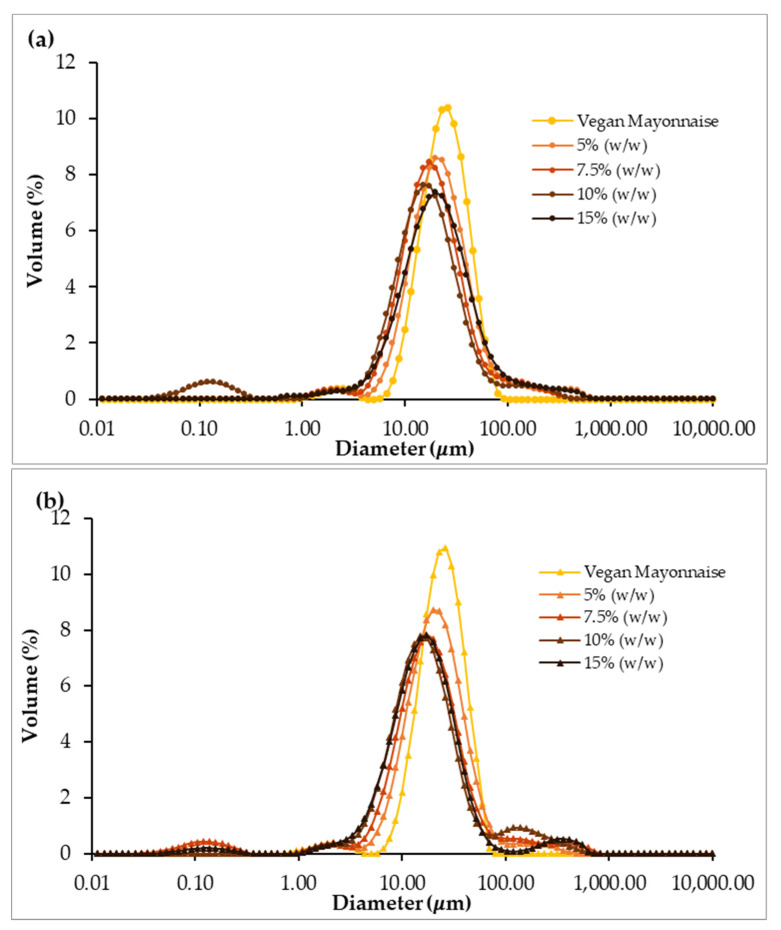
Comparison of droplet size distribution of emulsions with 5, 7.5, 10 and 15% (*w*/*w*) of miso paste and standard emulsion, measured 1 (**a**) and 30 (**b**) days after emulsion preparation.

**Figure 9 foods-12-01362-f009:**
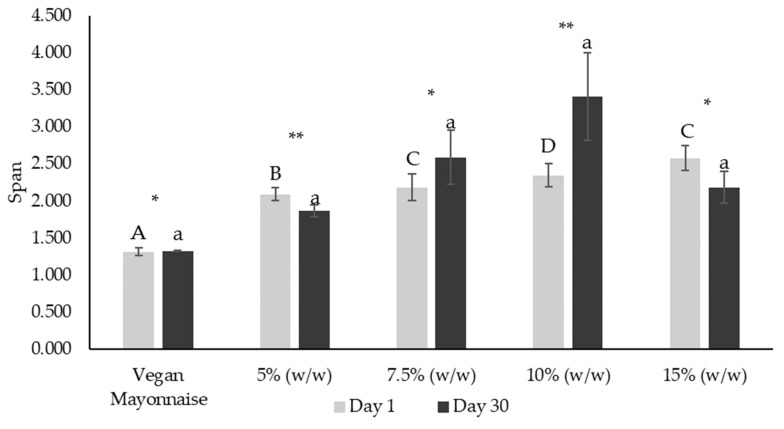
Span parameter from droplet size distribution of emulsions with 5, 7.5, 10 and 15% (*w*/*w*) of miso paste and standard emulsion, measured 1 and 30 days after emulsion preparation. Different letters indicate significantly different results (*p* < 0.05) according to Tukey test. One asterisk and two asterisks represent non significantly (*p* > 0.05) and significantly (*p* < 0.05) different results, respectively, according to the *t*-Test.

**Figure 10 foods-12-01362-f010:**
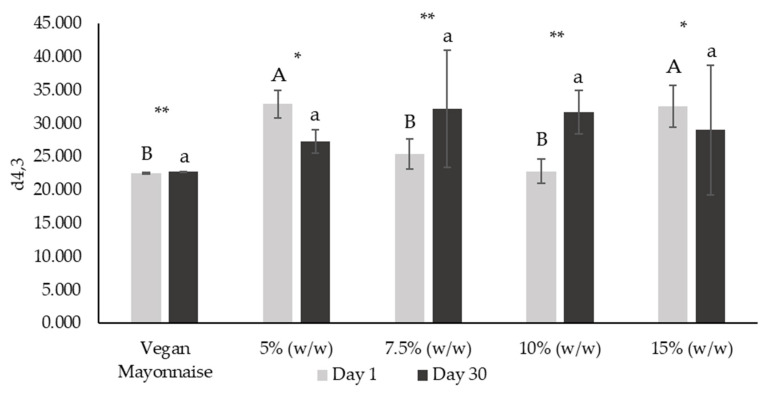
De Brouckere diameter (d_4,3_) from droplet size distribution of emulsions with 5, 7.5, 10 and 15% (*w*/*w*) of miso paste and standard emulsion, measured 1 and 30 days after emulsion preparation. Different letters indicate significantly different results (*p* < 0.05) according to Tukey test. One asterisk and two asterisks represent non significantly (*p* > 0.05) and significantly (*p* < 0.05) different results, respectively, according to the *t*-Test.

**Figure 11 foods-12-01362-f011:**
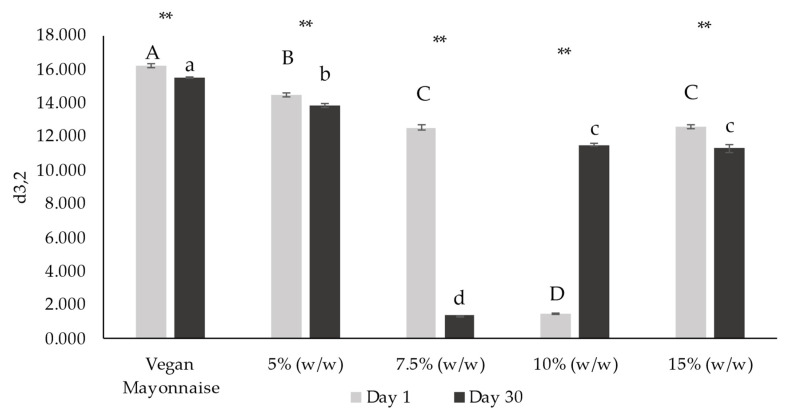
Sauter diameter (d_3,2_) from droplet size distribution of emulsions with 5, 7.5, 10 and 15% (*w*/*w*) of miso paste and standard emulsion, measured 1 and 30 days after emulsion preparation. Different letters indicate significantly different results (*p* < 0.05) according to Tukey test. Two asterisks represent non significantly (*p* > 0.05) and significantly (*p* < 0.05) different results, respectively, according to the *t*-Test.

**Figure 12 foods-12-01362-f012:**
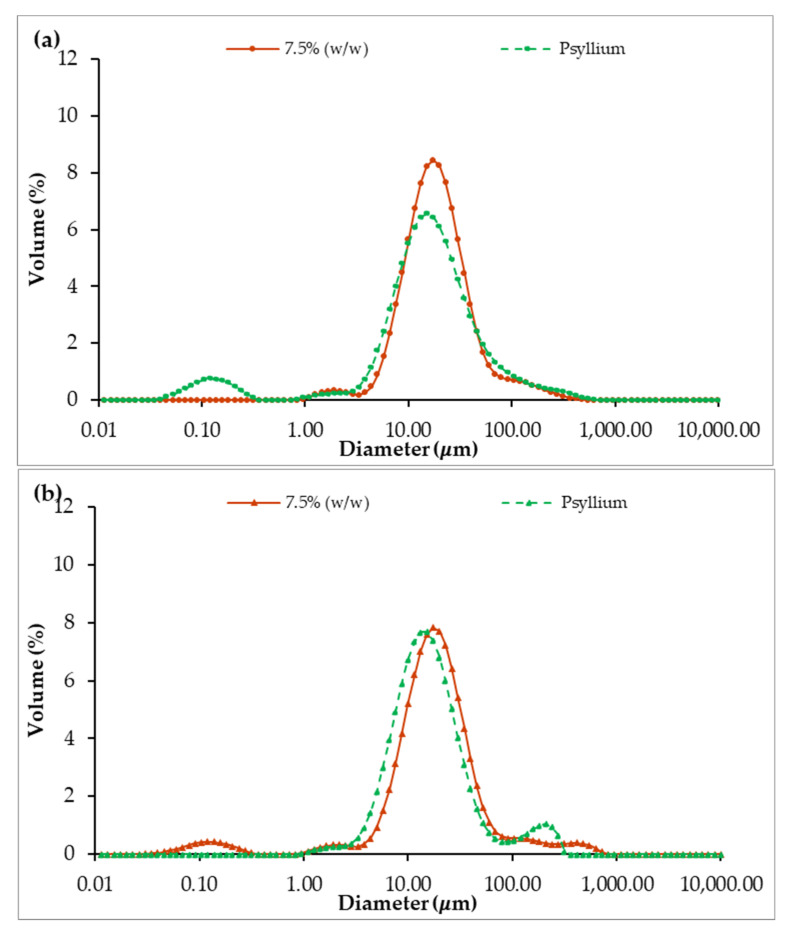
Comparison of droplet size distribution of emulsions with 7.5% (*w*/*w*) of miso paste, one of them with 0.1% (*w*/*w*) of psyllium husk, measured 1 (**a**) and 30 (**b**) days after emulsion preparation.

**Figure 13 foods-12-01362-f013:**
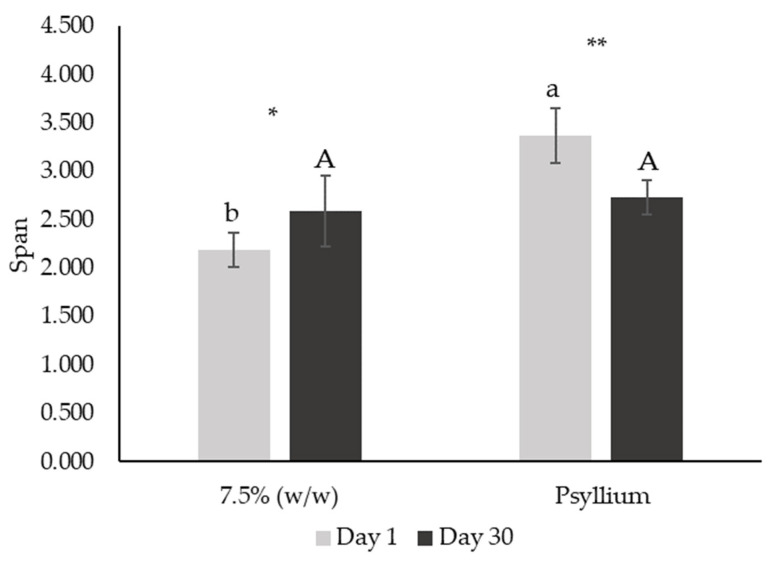
Span parameter from droplet size distribution of emulsions with 7.5% (*w*/*w*) of miso paste, one of them with 0.1% (*w*/*w*) of psyllium husk, measured 1 (a) and 30 (b) days after emulsion preparation. Different letters or two asterisks indicate significantly different results (*p* < 0.05) according to *t*-test. One asterisk indicates that samples did not present significantly different results (*p* > 0.05) according to *t*-test.

**Figure 14 foods-12-01362-f014:**
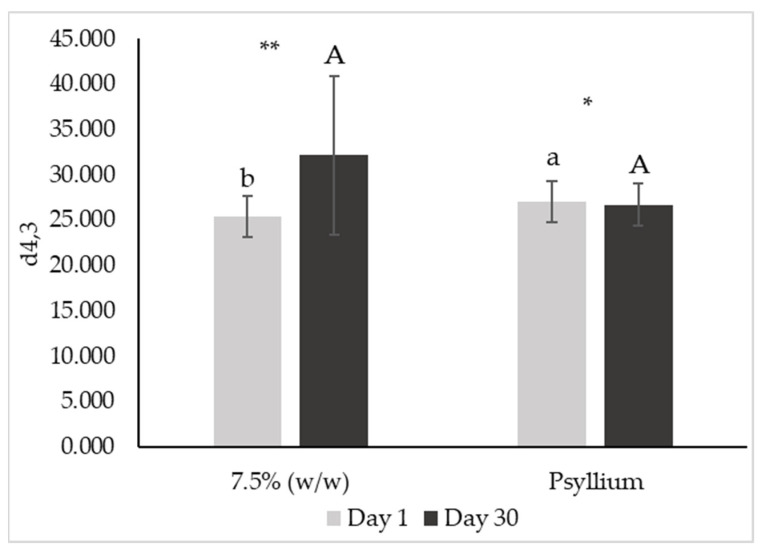
De Brouckere diameter (d_4,3_) from droplet size distribution of emulsions with 7.5% (*w*/*w*) of miso paste, one of them with 0.1% (*w*/*w*) of psyllium husk, measured 1 (a) and 30 (b) days after emulsion preparation. Different letters or two asterisks indicate significantly different results (*p* < 0.05) according to *t*-test. One asterisk indicates that samples did not present significantly different results (*p* > 0.05) according to *t*-test.

**Figure 15 foods-12-01362-f015:**
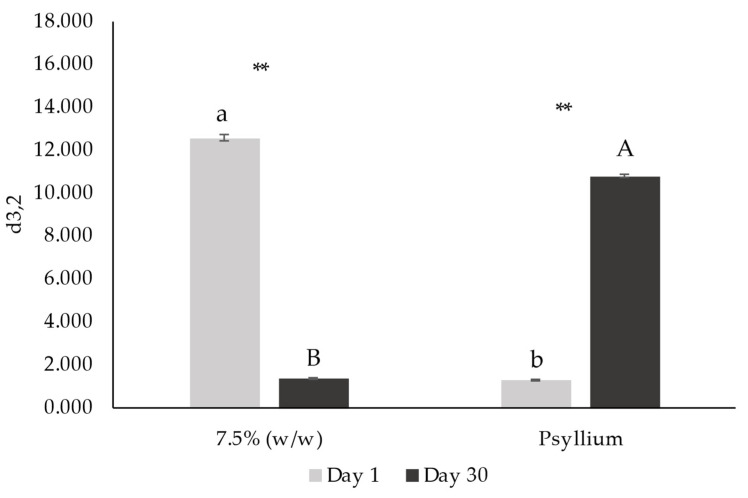
Sauter diameter (d_3,2_) from droplet size distribution of emulsions with 7.5% (*w*/*w*) of miso paste, one of them with 0.1% (*w*/*w*) of psyllium husk, measured 1 (a) and 30 (b) days after emulsion preparation. Different letters or two asterisks indicate significantly different results (*p* < 0.05) according to *t*-test. Two asterisks indicate that samples presented significantly different results (*p* > 0.05) according to *t*-test.

**Figure 16 foods-12-01362-f016:**
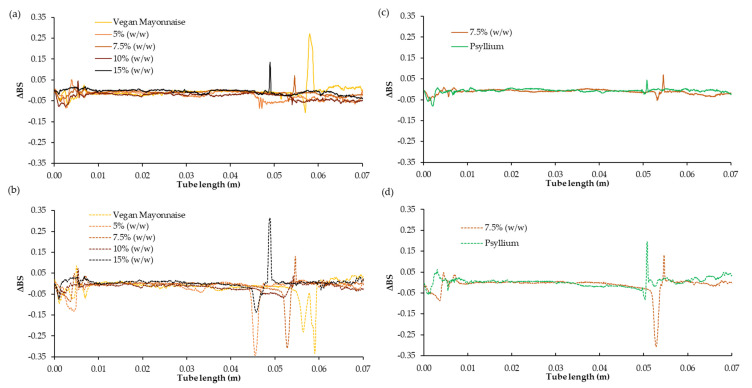
Changes in backscattering profiles (ΔBS) as a function of sample height of emulsions with 5, 7.5, 10 and 15% (*w*/*w*) of miso paste and standard emulsion, measured 1 day (**a**) and 30 days (**b**) after emulsion preparation. ΔBS profiles for 7.5% (*w*/*w*) of miso paste, one of them with 0.1% (*w*/*w*) of psyllium husk, measured 1 day (**c**) and 30 days (**d**) after emulsion preparation are also shown.

**Figure 17 foods-12-01362-f017:**
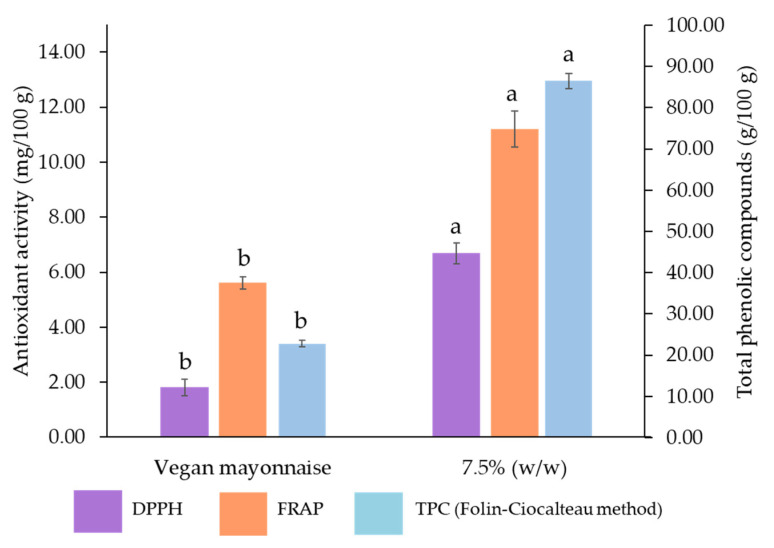
Total phenolic compounds (TPC) and antioxidant capacity (FRAP and DPPH methods) of standard and 7.5% (*w*/*w*) miso emulsions, expressed as mg of gallic acid equivalents per 100 g of emulsion. Different letters indicate significantly different results (*p* < 0.05) according to *t*-test.

**Table 1 foods-12-01362-t001:** Food pathogens and contaminant microorganisms used to test antimicrobial activity of grass pea sweet miso by drop test.

Group	Microorganism	Library Code	Optimal Growth Conditions	
			Nutritive Medium	Temperature
Food pathogens	*Listeria innocua*	BISA 3008	Blood Heart Infusion (BHI)	38 °C
	*Staphylococcus aureus*	BISA 3966		
	*Escherichia coli*	BISA 3967		
	*Salmonella* Typhimurium	BISA 3969		
	*Bacillus cereus*	BISA 4043		25 °C
Typical Mendes Gonçalves’ contaminants	*Lactiplantibacillus plantarum*	BISA 4386	De Man, Rogosa and Sharpe (MRS)	
	*Zygosacharomyces parabaillii*	ISA 1307	Yeast extract, peptone, dextrose and fructose (YPDF)	
	*Pichia membranifaciens*	ISA 1454		
	*Debaryomyces hansenii*	ISA 1509		
	*Zygosacharomyces bisporus*	ISA 2187		
	*Candida stelata*	ISA 2339		
	*Zygosacharomyces baillii*	ISA 2422		

**Table 2 foods-12-01362-t002:** Microbial analyses and methodologies performed for evaluating the microbial stability of emulsions.

Analysis	Methodology
Total mesophiles quantification	ISO 6610
Total molds and yeasts quantification	NP 3277-1
Quantification of total lactic acid bacteria	ISO 16649-2
Quantification of *Escherichia coli*	ISO 6888
Detection of *Listeria monocytogenes*	ISO 11290
Detection of *Salmonella* sp.	ISO 6579

**Table 3 foods-12-01362-t003:** Chemical composition and bioactivities of grass pea sweet miso and of traditional soybean miso.

Parameters	Wet Basis	
	Grass Pea Sweet Miso	Soybean Miso
Dry matter (g/100 g)	43.94 ± 0.36	54.78 [47]
Crude fat (g/100 g)	0.13 ± 0.04	11.00 [46]
Crude protein (g/100 g)	6.16 ± 0.17	11.98 [47]
Iron (mg/100 g)	1.45 ± 0.06	4.00 [46]
Calcium (mg/100 g)	43.75 ± 6.70	150.00 [46]
Sodium (g/100 g)	4.17 ± 1.00	4.30 [46]
Sodium chloride (g/100 g)	10.60 *	12.12 [47]
Total phenolic compounds (g of GAE/100 g)	0.68 ± 0.08	1.41–4.26 [48]
Antioxidant activity (FRAP assay, mg of GAE/100 g)	16.22 ± 0.13	0.60–1.35 [48]
Antioxidant activity (DPPH assay, mg of GAE/100 g)	5.29 ± 0.05	0.68–1.93 [48]

* Sodium chloride content was determined by multiplying the total sodium content by a conversion factor of 2.54 [28].

**Table 4 foods-12-01362-t004:** Parameters (G′ at 1 Hz and G^0^_N_) from mechanical spectra of standard and miso emulsions. Different letters indicate significantly different results (*p* < 0.05) according to Tukey test. Different amounts of asterisks indicate significantly different results (*p* < 0.05) according to *t*-test.

Emulsions	G′ at 1 Hz (10^2^ Pa)	Plateau Modulus, G^0^_N_ (10^2^ Pa)
Vegan mayonnaise	4.86 ± 0.21 ^d^	3.98 ± 0.18 ^d^
5% (*w*/*w*) miso	11.82 ± 0.92 ^c^	9.48 ± 0.65 ^c^
7.5% (*w*/*w*) miso	14.93 ± 0.44 ^ab,^ *	12.09 ± 0.35 ^ab,^ *
10% (*w*/*w*) miso	13.57 ± 1.03 ^bc^	10.88 ± 0.88 ^bc^
15% (*w*/*w*) miso	16.08 ± 1.06 ^a^	12.83 ± 0.86 ^a^
Psyllium	12.51 ± 0.49 **	9.27 ± 0.14 **

**Table 5 foods-12-01362-t005:** Parameters (*η_0_*, *k* and *m*) and adjusted R^2^ from Williamson’s model adjusted to the flow curves of standard and miso emulsions. Different letters indicate significantly different results (*p* < 0.05) according to Tukey test. Different amounts of asterisks indicate significantly different results (*p* < 0.05) according to *t*-test.

Emulsions	*η_0_* (10^3^ Pa.s)	*k* (10^3^ Pa.s)	*m* (Dimensionless)	Adjusted R^2^
Vegan mayonnaise	13.31 ± 0.50 ^b^	1.39 ± 0.18	0.982 ± 0.008	0.9971
5% (*w*/*w*) miso	31.96 ± 0.96 ^ab^	1.37 ± 0.07	1.016 ± 0.028	0.9965
7.5% (*w*/*w*) miso	40.85 ± 045 ^a,^ *	1.05 ± 0.07	1.000 ± 0.020	0.9967
10% (*w*/*w*) miso	33.69 ± 8.61 ^ab^	1.37 ± 0.32	0.976 ± 0.012	0.9869
15% (*w*/*w*) miso	41.89 ± 3.86 ^ab^	1.82 ± 1.67	0.971 ± 0.027	0.9947
Psyllium	28.61 ± 8.5 *	1.25 ± 0.32	0.972 ± 0.020	0.9917

**Table 6 foods-12-01362-t006:** Evolution of microbial counts in 7.5% (*w*/*w*) miso emulsion, from day 0 to day 30, after 30 days under refrigeration or under accelerated assay.

Microbial Study	Day 0	Day 30—Refrigeration	Day 30—Accelerated Assay
Total mesophiles (10^1^ CFU/g)	285.00 ± 162.64	410.00 ± 7.07	<1
Yeasts and molds (10^2^ CFU/g)	155.00 ± 35.36	56.00 ± 7.07	<1
Lactic acid bacteria (CFU/g)	<10	57.50 ± 3.54	<10
*E. coli* (CFU/mL)	<10	<10	<10
*Listeria monocytogenes* (in 25 g)	Absent	Absent	Absent
*Salmonella* sp. (in 25 g)	Absent	Absent	Absent

## Data Availability

Research data are available from the corresponding author upon reasonable request.

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
