# Peer review of "Impact of Grass Pea Sweet Miso Incorporation in Vegan Emulsions: Rheological, Nutritional and Bioactive Properties"

_foods, 2023, doi:10.3390/foods12071362_

Round 1
Reviewer 1 Report
This manuscript investigated the rheological, nutritional, and bioactive properties of grass pea sweet miso incorporation in vegan emulsions. The topic is interesting; however, the manuscript has several problems.
1. L 23; Mention the percentage of psyllium husk.
2. L 74-78; This paragraph does not need a reference. Also, what is the novelty of the current work with the previous one?
3. Provide some information about vegan diet in the introduction section.
4. Please check the reference style, for example: L 81.
5. L 88; Provide exact percentages of each protein content.
6. Section 2.2.3; Please specify the treatments (control, etc.). You can even use the table for better understanding.
7. Table 2 needs some references.
8. Sections 3.2.1 and 3.2.2; The results should be provided as a supplementary file.
9. Figures 11, 12, and 16; Check the standard deviation. This amount of standard deviation is not reasonable.
10. Please mention the best formulation in the conclusions section.
Author Response
Dear reviewer, we appreciate your kind words. We took your advice in consideration. All alterations in the text are highlighted in yellow for your convenience.
Point 1: L 23; Mention the percentage of psyllium husk.
Response 1: This information was added on line 22.
Point 2: L 74-78; This paragraph does not need a reference. Also, what is the novelty of the current work with the previous one?
Response 2: The reference was added since the development of the grass pea sweet miso is already published, making further information easily available for the readers. Our work consisted of incorporating the grass pea sweet miso previously developed by colleagues, using it as an ingredient in a more complex product.
Point 3: Provide some information about vegan diet in the introduction section.
Response 3: Since the two base products for this work are already vegan, the authors did not find necessary to thoroughly explore the vegan diet concept. However, some information was added.
Point 4: Please check the reference style, for example: L 81.
Response 4: The reference appears in square brackets as defined by Foods. Nevertheless, “Santos el al.” was added in the text as a way of speech, informing the reader that the grass pea miso was not developed in this work, but in a previous work, already published.
Point 5: L 88; Provide exact percentages of each protein content.
Response 5: The protein content was not analysed, as the protein’ manufacturers provide a specification sheet. For both ingredients, the protein content is >85% (w/w), as described on the manuscript.
Point 6: Section 2.2.3; Please specify the treatments (control, etc.). You can even use the table for better understanding.
Response 6: The developed emulsions did not vary in treatment. The variation consists of different levels of incorporation of grass pea sweet miso and psyllium husk. This information was clarified on lines 150-152: The control emulsion consisted of a faba bean and lupin emulsion without the addition of grass pea sweet miso or psyllium husk.
Point 7: Table 2 needs some references.
Response 7: Table 2 contains the reference to different Portuguese and international standards.
Point 8: Sections 3.2.1 and 3.2.2; The results should be provided as a supplementary file.
Response 8: These sections were added in the main document since they are important for the readers understanding of the developed work and course of action, justifying why the miso incorporation phase was selected, and why psyllium husk was added.
Point 9: Figures 11, 12, and 16; Check the standard deviation. This amount of standard deviation is not reasonable.
Response 9: The big standard error was derived from a replica that was very different from the others. This was removed, with the addition of a different replica, and the results changed accordingly.
Point 10: Please mention the best formulation in the conclusions section
Response 10: This information was added on lines 679-680: This incorporation was made on the most promising miso concentration, 7.5% (w/w).
Dear reviewer, once again, we appreciate all your feedback and took it in consideration to improve our manuscript. Best regards.
Reviewer 2 Report
This manuscript “Impact of Grass pea sweet miso incorporation in vegan emulsions: rheological, nutritional, and bioactive properties” was prepared not well. I think that it is not proper to publish before checking the full manuscript and make some modifications. Some comments are shown as follow.
1. Abstract need to be modified. Besides, Line 24 “a significantly higher content of phenolic compounds and antioxidant activity” has no compared objects. Besides, the potential application needs to indicate clearly.
2. Keyword “grass pea sweet miso” includes “Lathyrus sativus”.
3. Why analyze Impact of Grass pea sweet miso incorporation in vegan emulsions? There is no description about vegan emulsions.
4. In Line 82, what is the ratio? Volume or mass? This should be indicated clearly.
5. Line 211 “psyllium emulsions” contains an amount of water. How to observe them using SEM? Maybe use light microscopy, fluorescent microscopy or CLSM like previous reports about fish oil-in-water pickering emulsions structured with tea water-insoluble proteins/κ-carrageenan complexes.
6. the speed unit should be consistent. “g” or “rpm”?
7. In Table 3, what does “*” represent for ?
8. Part “3.2.1. Preliminary tests - Incorporation phase” should be deleted.
9. “as it prevented the release of exudate without further alterations on the emulsion’s texture and mouthfeel (results unpublished).” can be supplied as the attachment materials.
10. All the figures need to be improved. Besides, the upper letters in figures were different style, what is the difference between capital letter and small letter. These problems appeared in the other figures and tables. They should be indicated clearly.
11. The reference should be checked clearly. Some are not normative.
Author Response
Dear reviewer, we appreciate your words. We took your advice in consideration. All alterations in the text are highlighted in yellow for your convenience.
Point 1: Abstract need to be modified. Besides, Line 24 “a significantly higher content of phenolic compounds and antioxidant activity” has no compared objects. Besides, the potential application needs to indicate clearly.”
Response 1: The information from line 24 was cleared. The potential application of this work is described in lines 15-17.
Point 2: Keyword “grass pea sweet miso” includes “Lathyrus sativus”.
Response 2: We believe that the keywords correspond to different elements. Grass pea sweet miso is a novel food ingredient, not so well explored. On the other hand, Lathyrus sativus is a legume, which is the raw material for the miso production. However, it is not desirable that a key word should be just “sweet miso”, since there are several types of miso, to not be confused. However, since grass pea sweet miso is a novel ingredient, to use it as keyword will facilitate the search of readers.
Point 3: Why analyze Impact of Grass pea sweet miso incorporation in vegan emulsions? There is no description about vegan emulsions.
Response 3: These vegan emulsions are already described and published by Cabrita et. al (2022), one of our references. They are very similar to mayonnaise. The use of grass pea sweet miso as an ingredient of this emulsions is the novelty of our work, aiming to creative an innovative added-value product. For its complex composition, it is vital to assess the impact of grass pea sweet miso in the emulsions structure, that will suffer alterations.
Point 4: In Line 82, what is the ratio? Volume or mass? This should be indicated clearly.
Response 4: This information was clarified
Point 5: Line 211 “psyllium emulsions” contains an amount of water. How to observe them using SEM? Maybe use light microscopy, fluorescent microscopy or CLSM like previous reports about fish oil-in-water pickering emulsions structured with tea water-insoluble proteins/κ-carrageenan complexes.
Response 5: All the emulsions contain water, with or without psyllium. Hence, their observation in SEM is possible.
Point 6: the speed unit should be consistent. “g” or “rpm”?
Response 6: The speed units are now all in rpm (changed on line 240), since it is not possible to obtain UltraTurrax’s speed in g.
Point 7: In Table 3, what does “*” represent for ?
Response 7: The meaning of * was added to Table 3.
Point 8: Part “3.2.1. Preliminary tests - Incorporation phase” should be deleted.
Response 8: This section was added in the main document since it is important for the readers understanding of the developed work and course of action, justifying why the miso incorporation phase was selected.
Point 9: as it prevented the release of exudate without further alterations on the emulsion’s texture and mouthfeel (results unpublished).” can be supplied as the attachment materials.
Response 9: This phrase was altered to “(pass/fail decisions based on like/dislike on simple taste tests, results unpublished), as the results derived from empirical sensory analysis on the lab”
Point 10: All the figures need to be improved. Besides, the upper letters in figures were different style, what is the difference between capital letter and small letter. These problems appeared in the other figures and tables. They should be indicated clearly.
Response 10: As indicated in figure and table’ captions, different letters represent significantly different results according to Tukey’s or t-tests. Capital and small letters are used to differentiate different statistic tests on the same figure. For example, Figure 4, firmness results were not compared to adhesiveness results. Hence, using the letter “a” for two sample for the two parameters could be confusing for some readers, indicating that we compared firmness and adhesiveness, which would not be correct.
Point 11: The reference should be checked clearly. Some are not normative.
Response 11: A reference manager software was used to check all references, they should now be correct. In some cases, a name and year are referred for phrasing purposes, but the correct reference inside brackets is presented as well.
Dear reviewer, once again, we appreciate all your feedback and took it in consideration to improve our manuscript. Best regards.

Reviewer 3 Report
To whom it may concern,
The current manuscript entitled "Impact of Grass pea sweet miso incorporation in vegan emulsions: rheological, nutritional, and bioactive properties" is about the incorporation of grass pea sweet miso in vegan emulsions with aims to increase the use of grass pea in the Portuguese diet, as well as to develop innovative emulsions aligned with consumer’s trends. The subject is interestiong but it needs some improvements especially in the discussion section, the problem is that the authors only representing their results but no discussion about them. My specific comments are given in PDF file as attached.

Author Response
Dear reviewer, we appreciate your words. We took your advice in consideration. All alterations in the text are highlighted in yellow for your convenience. Some of your comments were addressed on this letter when further explanation was necessary.
Point 1: Suggest to change the title to: A vegan approach of grass pea sweet miso incorporation into emulsions: Rheological, nutritional and bioactivity perspectives.
Response 1: We do not agree with this change, as it would completely alter the meaning of our work. Grass pea sweet miso was added to an already optimized vegan emulsion, as a way to increase the use of grass pea sweet miso in Portuguese diet. This title suggests that grass pea sweet miso is already used in traditional emulsions, and that we are only making it vegan, which does not correspond to the truth.
Point 2: high or minimum". Delete either of them.
Response 2: The phrase “Even though miso contains high minimum amounts of salt in its composition…” means that miso must have minimum amount of salt in it composition, and that, comparing to other food products, this amount is rather high, thus the expression “high minimum amount”.
Point 3: Is there any studies by now about the incorporation of grass pea sweet miso into emulsions. Please mention.
Response 3: Our study is original and innovative, since grass pea miso has not been previously incorporated in vegan emulsions. Our research group has, however, some unpublished studies regarding incorporation in other food products such as cookies and cracker.
Point 4: In addition, or furthermore or moreover. "Also" is not normally used in paper writing.
Response 4: This expression was changed to “Furthermore”.
Point 5: Please discuss the obtained results. You are only reporting your results with no discussion. This section is results and discussion. So, it means that beside reporting your results, you should discuss and elaborate more.
Response 5: The results are presented and analysed immediately after, as the table is presented in line 279, and results are discussed in line 280.
Point 6: Why did you use the previous results and didnt anlyze by yourself.
Response 6: The aim of our study was to incorporate grass pea sweet miso, and not another one, on vegan emulsions. Hence, no analyses were performed using soybean miso. Therefore, this analysis based on literature was added to enrich the article and to enable the comparison of our innovative ingredient with a traditional and widely studied and consumed fermented product, soybean miso.
Point 7: Why is standard error so big !
Response 7: The big standard error was derived from a replica that was very different from the others. This was removed, with the addition of a different replica, and the results changed accordingly.
Point 8: The number of Figures are too many. You can combine them, for example, Figures 4 and 5 can be combined. The same thing for other Figures.
Response 8: The number of figures aimed to improve their resolution. However, the figures were combined when fit.
Dear reviewer, once again, we appreciate all your feedback and took it in consideration to improve our manuscript. Best regards.

Round 2
Reviewer 1 Report
Thanks for considering my suggestions, however, some of them still remain! For example,
1. L 89, The reference should be removed. Because the sentence starts with "Our goal..." and does not need a reference!
2. "Santos et al. (2021) [9]" or "Cabrita et al. (2022) [19]” should change to “Santos et al. [9]" and "Cabrita et al. . [19]". The entire manuscript should be revised.
3. All the references in Table 2 should be mentioned in the "Reference" section.
Reviewer 2 Report
This manuscript was not well prepared. It must be revised carefully. Some are listed as follow.
1. The author may not understand question “Why analyze Impact of Grass pea sweet miso incorporation in vegan emulsions? There is no description about vegan emulsions.”. This need to describe clearly in the introduction. If you do not tell readers, how to let them understand your purpose? This must tell clearly.
2. “All the emulsions contain water, with or without psyllium. Hence, their observation in SEM is possible.”. SEM is usually used to analyze solid rather than solution. How do you test emulsions? Is there some special treatment? Please describe it clearly in the method.
3. All the figures are not revised. They must be improved. They are not satisfied to the style of publication. There are no scale marks in all the figures. Besides, the significant labels should be consistent in all the figures. Do not use different label styles to show the significance, for example, a, x, A….
4. As the author said, Part “3.2.1. Preliminary tests - Incorporation phase” was added in the main document since it is important for the readers understanding of the developed work and course of action, justifying why the miso incorporation phase was selected. Then the data should be provided.
5. References are not revised well. Some have DOI. Some do not show DOI such as reference 6, 32,33... Besides, some are lack of pages , for example, Ref 39,45,45,60. Please check all the reference.
Author Response
Dear reviewer, once again, we appreciate your words. As before, your advice was taken in consideration to improve the manuscript. Alterations in the text are highlighted in yellow and track changes mode was used for your convenience.
Point 1: The author may not understand question “Why analyze Impact of Grass pea sweet miso incorporation in vegan emulsions? There is no description about vegan emulsions.”. This need to describe clearly in the introduction. If you do not tell readers, how to let them understand your purpose? This must tell clearly.
Response 1: After review #1, some information about vegan emulsions was included in the introduction. However, some more information was added, justifying the incorporation of grass pea sweet miso in this food matrix.
Point 2: “All the emulsions contain water, with or without psyllium. Hence, their observation in SEM is possible.”. SEM is usually used to analyze solid rather than solution. How do you test emulsions? Is there some special treatment? Please describe it clearly in the method.
Response 2: In our research group we have been using this tabletop scanning electron microscope (TM3000, Hitachi, Tokyo, Japan), that works under low vacuum (30–50 Pa) and low temperature (- 4 ËšC), enabling the analysis of emulsions, with no thorough adaptations to this type of samples. Furthermore, we have already published results with this methodology, including in Foods. However, considering this manuscript includes several other methodologies, and that SEM is not so vital for the quality and relevance of the results, we agree to remove them.
Point 3: All the figures are not revised. They must be improved. They are not satisfied to the style of publication. There are no scale marks in all the figures. Besides, the significant labels should be consistent in all the figures. Do not use different label styles to show the significance, for example, a, x, A….
Response 3: The scale marks were included when needed. Moreover, statistical labels were all changed to lowercase letters in most figures and tables. However, some exceptions were made. A) Tables 4 and 5: two statistical tests were made, Tukey test and t-test. Hence, using only lowercase letters could be confusing, thus the usage of asterisks to indicate t-test results. This was clarified in the capture. B) Figures 9, 10, 11, 13, 14 and 15. Taking figure 9 as example, since the motivation is the same for figures 9-11: three different statistical tests were used in this figure. A Tukey test was used to compare the Span of the 5 emulsions on day 1; a second Tukey test was used to compare the Span of the 5 emulsions on day 30; and a t-test was used to compare day 1 and day 30 of each emulsion.
Taking figure 13 as example, since the motivation is the same for figures 13-15: three different statistical tests were used in this figure. A t-test was used to compare the Span of the 2 emulsions on day 1; a second t-test was used to compare the Span of the 2 emulsions on day 30; and a third t-test was used to compare day 1 and day 30 of each emulsion. Hence, using only lowercase letters would certainly confuse the reader, that would not be able to make any conclusions just by observing the figures, forcing them to read the text for better understanding. Since figures are supposed to be self-explanatory, the authors would appreciate that the use of both lower and upper case letters, as well as asterisks, would be accepted.
Point 4: As the author said, Part “3.2.1. Preliminary tests - Incorporation phase” was added in the main document since it is important for the readers understanding of the developed work and course of action, justifying why the miso incorporation phase was selected. Then the data should be provided.
Response 4: The inclusion of this data would extend an already long manuscript. The main conclusions are provided on lines 363-366, since all the results obtained pointed the same: the addition of miso after protein hydration enabled the best results, with emulsions closer to the target. Since this was only tested for a preliminary formulation, the authors don’t consider these results relevant for publishing. Nevertheless, we consider important that it is clear in the manuscript that the incorporation phase was not decided by change, but by scientific testing and decision making, although not relevant for publishing.
Point 5: References are not revised well. Some have DOI. Some do not show DOI such as reference 6, 32,33... Besides, some are lack of pages , for example, Ref 39,45,45,60. Please check all the reference.
Response 5: The references were revised, and information added when necessary. Reference 33 does not have a DOI, so we included ISBN. For reference 55, no DOI nor ISBN were found. Since these alterations were made using automatic software, it was not possible to use yellow highlighting on the alterations. However, track changes mode was turned on.
Dear reviewer, once again, we appreciate all your feedback and took it in consideration to improve our manuscript. Best regards.
